# The Role of Consumption of Molybdenum Biofortified Crops in Bone Homeostasis and Healthy Aging

**DOI:** 10.3390/nu15041022

**Published:** 2023-02-17

**Authors:** Sonya Vasto, Davide Baldassano, Leo Sabatino, Rosalia Caldarella, Luigi Di Rosa, Sara Baldassano

**Affiliations:** 1Euro-Mediterranean Institutes of Science and Technology (IEMEST), 90139 Palermo, Italy; 2Department of Biological, Chemical and Pharmaceutical Sciences and Technologies, University of Palermo, 90133 Palermo, Italy; 3Department of Promoting Health, Maternal-Infant, Excellence and Internal and Specialized Medicine (ProMISE) G. D’Alessandro, University of Palermo, 90127 Palermo, Italy; 4Dipartimento Scienze Agrarie, Alimentari e Forestali (SAAF), University of Palermo, Viale delle Scienze, Ed. 5, 90128 Palermo, Italy; 5Department of Laboratory Medicine, “P. Giaccone” University Hospital, 90127 Palermo, Italy

**Keywords:** nutrition, human health, *Lactuca sativa* L., micronutrients, supplementation, gut-bone axis

## Abstract

Osteoporosis is a chronic disease and public health issue in aging populations. Inadequate intake of micronutrients increases the risk of bone loss during an adult’s lifespan and therefore of osteoporosis. The aim of the study was to analyze the effects of consumption of biofortified crops with the micronutrient molybdenum (Mo) on bone remodeling and metabolism in a population of adults and seniors. The trial enrolled 42 senior and 42 adult people randomly divided into three groups that consumed lettuce biofortified with molybdenum (Mo-biofortified group) or without biofortification (control group) or molybdenum in a tablet (Mo-tablet group) for 12 days. We chose an experimental period of 12 days because the bone remodeling marker levels are influenced in the short term. Therefore, a period of 12 days allows us to determine if there are changes in the indicators. Blood samples, obtained at time zero and at the end of the study, were compared within the groups adults and seniors for the markers of bone resorption, *C*-terminal telopeptide (CTX) and bone formation osteocalcin, along with the markers of bone metabolism, parathyroid hormone (PTH), calcitonin, albumin-adjusted calcium, vitamin D, phosphate and potassium. Consumption of a Mo tablet did not affect bone metabolism in the study. Consumption of Mo-biofortified lettuce significantly reduced levels of CTX and PTH and increased vitamin D in adults and seniors while levels of osteocalcin, calcitonin, calcium, potassium and phosphate were not affected. The study opens up new considerations about the role of nutrition and supplementation in the prevention of chronic diseases in middle-aged and older adults. Consumption of Mo-biofortified lettuce positively impacts bone metabolism in middle-aged and older adults through reduced bone resorption and improved bone metabolism while supplementation of Mo tablets did not affect bone remodeling or metabolism. Therefore, Mo-biofortified lettuce may be used as a nutrition intervention to improve bone homeostasis and prevent the occurrence of osteoporosis in the elderly.

## 1. Introduction

Aging is associated with a decline of functional capacity and the inability to adapt to stress responses. The aging condition itself impacts the physiological performance of the human body and increases the susceptibility to age-related diseases and death. Common conditions in older age are reduction in metabolism (e.g., glucose, lipid, bone, skeletal muscle) and reduction in hearing and eyesight [1]. There is general bone and central nervous system deterioration, although new pieces of evidence suggest that the aging process can be delayed and modifiable [2]. Osteoporosis is an age-related disease that results in reduced bone mass, disruption of bone architecture and increased risk of fractures [3]. At a molecular biology level, deregulation in signals and pathways of Wnt, RUNX2, RANKL, OPG, Osx, CBFB, BMP-2, FoxOs, Nrf2, Gsα and sclerostin leads to osteoporosis [4]. For example, changes at the genetic and epigenetic levels may lower or over-activate the Wnt signaling pathway and thereby cause osteoporosis. In addition, the mutations in LRP5 cause osteoporosis-pseudoglioma syndrome (OPPG) autosomal disorder [5]. Thus, the Wnt signaling pathway is a candidate for therapeutic intervention for osteoporosis [6]. There are several risk factors accounting for osteoporosis development; some are unmodifiable, such as age, gender, menopause, previous fractures, family history [1] and medication, and some can be amended, such as lifestyle issues, including diet [7] exercise, smoking and alcohol [8]. Adequate nutrition, in terms of macronutrients and micronutrients, can have a central role in osteoporosis prevention and treatment.

Micronutrient needs for optimizing well-being in theory can be met by a healthy diet. However, most of the population does not have such a diet, especially during times when their well-being is challenged [3,7,9,10,11]. Among different agronomic practices [12,13,14], agronomic biofortification is recognized as one of the most promising, efficient, sustainable and cost-effective strategies for overcoming mineral malnutrition [15,16,17,18,19,20]. Therefore, biofortified food could be considered a good option for meeting nutrient needs during different stages of life and aging [21]. It could be easily supported by sustainable, safe agriculture [22] and prescribed together with a nutritious diet [7,10,23].

Molybdenum (Mo) is a micronutrient essential for many processes in the body. A Mo cofactor activates essential enzymes, such as sulfite oxidase, aldehyde oxidase, xanthine oxidase and mitochondrial amidoxime reducing component (mARC) [24]. It is contained in soil and transferred to us when we consume plants, legumes and grains, as well as animals that fed on those [25]. Remarkably, molybdenum is rapidly excreted in urine [26]. The recommended dietary allowance (RDA) is 45 μg/day and the biomonitoring equivalent (BE) molybdenum values associated with toxicity in serum and urine are 31 μg/L and 7500 μg/L, respectively. These values were calculated by considering the tolerable daily intake exposure guidance values and the reference dose [27]. In one study, a nutritional intervention with Mo-biofortified lettuce ameliorated iron metabolism in healthy adult individuals [28]. Furthermore, in another, the ingestion of Mo-enriched lettuce improved glucose homeostasis by increasing, within the physiological range, endogenous gut peptide levels of peptide YY (PYY) and gastric inhibitory polypeptide (GIP) [29]. Gut peptides are released following food intake, and they are able to influence bone remodeling [30,31] and inhibit bone resorption [32]. Hence, we hypothesized that bone turnover would be influenced by a short-term nutritional intervention with biofortified molybdenum lettuce and it would impact the essential regulators of bone metabolism such as calciotropic hormones and markers of calcium metabolism. Therefore, this nutritional research aimed to investigate whether 12 days of consumption of Mo-biofortified lettuce could affect bone turnover markers in a population of adults and seniors. Furthermore, we compared the obtained results to a cohort of people supplemented with a molybdenum tablet in their diet.

## 2. Materials and Methods

### 2.1. Research Design of the Study

This research is part of the larger project, Nutri-Mo-Food, that is registered at Clinicaltrials.gov (accessed on 12 February 2023) with the approbation number NCT04985240. The protocol, conducted in accordance with the Declaration of Helsinki, was approved by the Palermo University Hospital Ethics Committee, and the approbation number is 2/2020 AIFA CE 150109. From an initial 95 subjects (47 adults age range 23–54 and 48 seniors age range 55–73), on the basis of the eligibility criteria, 11 subjects were excluded (5 in the adults and 6 in the senior group), and 84 subjects (42 adults comprising 25 males and 17 females and 42 seniors comprising 24 males and 18 females) were randomized into the control group, the Mo-tablet and the biofortified group for a total of 14 subjects in each group of adults and seniors (Figure 1). The participants at the study, the medical staff and the investigator staff were blinded to the allocation during the whole period of data collection. The investigators were also blinded during the sample assessment and data analysis. The control group received 100 g of lettuce and the intervention group received 100 g of Mo-fortified lettuce to consume every day for a total of 12 days. The molybdenum supplementation tablet (150 µg/tablet) was purchased from Zein-pharma (Germany). The Mo-tablet group received one tablet of Mo to consume daily for a total of 12 days. Samples of blood were collected before (T0) and at the end (T1) of the trial, after 12 days (Figure 2).

Exclusion and inclusion criteria are reported in Table 1.

In the first visit, the subjects underwent anthropometric measurement and completed a habitual dietary intake assessment [23]. Participants were recommended not to change their habits and diet during the study period, which was reiterated throughout the study. Body composition (fat and lean mass) was measured by electrical bioimpedance measurements (InBody 320 Body Composition Analyzer). Barefoot standing height and body weight were measured by using a wall-mounted stadiometer (Gima 27088, Italy) and an electronic scale (Gima 27335) [30]. Body mass index (BMI) was reported as weight, measured in kilograms, per standing height, measured as meters squared.

### 2.2. Sample Size

We conducted an a priori power calculation based on previous studies with hematological parameters [29,30,33], and by setting a level of statistical significance for α of 5% and a probability for β of 20%, we estimated a sample size of eight participants was required in each group. We included 14 subjects in the study to reduce the type-two error risks and to enhance the power of evaluation for the secondary outcomes.

### 2.3. Analysis of the Bone Markers and Metabolism

The samples of blood were collected at the same time of day, in the morning between 07:00 and 08:00 a.m., in order to reduce the circadian variation. The samples were collected after overnight fasting from 8:00 p.m. the day before. The blood was dispensed in EDTA tubes for plasma collection (centrifugation at 1509× *g* for 10 min at 4 °C) while it was dispensed in VACUETTE ^®^ serum tubes and allowed to clot for 30 min at room temperature before centrifugation for serum collection. Samples were analyzed by using a method that is FDA cleared and CE marked. Specifically, electro-chemiluminescence immunoassay (ECLIA) (Roche Diagnostics, Burgess Hill, UK) on a Cobas e601 analyzer for dosage in sample serum of CTX, cat. number 11972308122, vitamin D, cat. number 07464215190, osteocalcin, cat. number 12149133122, PTH, cat. number 11972103122 and calcitonin, cat. number 06445853190. We used Roche COBAS c501 and standard commercial assays supplied by Roche Diagnostics for dosage in plasma samples of calcium, cat. number 106443, albumin, cat. number 05166861, phosphate, cat. number 05171377 and potassium, cat. number 11208764202. The total calcium concentrations may change independently of the ionized calcium concentration because of fluctuations in albumin protein concentrations. Therefore, the concentration of calcium was corrected using the following equation: (−0.8 × ([albumin] − 4)) + [total Ca] to give an albumin-adjusted calcium (aa calcium) value [34].

### 2.4. Statistical Analysis

Student’s t tests were used to analyze and compare the characteristics at baseline of the study groups, and one-way ANOVA followed by Tukey’s post-test were used to compare differences between baseline and time 1. *p* ≤ 0.05 was considered statistically significant. Data are presented as means ± standard deviations.

## 3. Results

The anthropometric data of the participants enrolled in the study are shown in Table 2 for the beginning (baseline), before starting the nutritional intervention. There were no significant changes in BMI, body weight or lean and fat body masses within the groups during the nutritional protocol intervention.

### 3.1. Mo-Biofortified Lettuce and Markers of Bone Remodeling

The nutritional intervention with Mo-biofortified lettuce significantly affected circulating concentrations of CTX, the marker of bone resorption. In fact, in the Mo-biofortified group, plasma concentrations of CTX were significantly reduced after 12 days compared to baseline (Figure 3A). In the adult group, the serum concentration of CTX at the end of the nutritional intervention decreased by 52% from baseline (0.47 ± 0.1 vs. 0.24 ± 0.1 µg/L). In the senior group, the serum concentration of CTX at the end of the nutritional intervention decreased by 58% from baseline (0.48 ± 0.2 vs. 0.28 ± 0.1 µg/L). The CTX concentration detected was within the physiological range in both adult and senior groups. Osteocalcin, the marker of bone formation, was not affected by the nutritional intervention with Mo-biofortified lettuce after 12 days compared to baseline (adults: 22.5 ± 8 vs. 21.6 ± 6 µg/L; seniors 21.1± 7.2 vs. 20.7 ± 5.7 µg/L) (Figure 3B). In the control group of both adults and seniors who consumed lettuce for 12 days, there were no differences in the bone remodeling markers compared to baseline at the end of the intervention (Figure 3). Moreover, consumption of a Mo-tablet for 12 days did not affect markers of bone remodeling in adults or seniors within the group with respect to the control group (Figure 3). The Mo-tablet group was supplemented with a tablet of Mo containing 150 µg of sodium molybdate and the Mo-biofortified group was supplemented with 100 g a day of molybdenum-enriched lettuce containing 8 mg of sodium molybdate.

The comparisons between the groups (control vs. Mo-biofortified and Mo-tablet vs. Mo-biofortified) showed significant changes in CTX but not in the osteocalcin concentration after 12 days of nutritional intervention in the Mo-biofortified group with respect to the control and Mo-tablet groups in both adults and seniors (Figure 3).

### 3.2. Mo-Biofortified Lettuce and Markers of Bone Metabolism 

The nutritional intervention with Mo-biofortified lettuce significantly affected the marker of bone metabolism, PTH. In the Mo-biofortified group, PTH concentrations decreased with time. Specifically, PTH at baseline was significantly different from day 12 (adults: 36.5 ± 8.3 vs. 22.3 ± 6.7 ng/L; seniors: 42.4 ± 8 vs. 32.8 ± 5.4 ng/L) (Figure 4A). The PTH levels at the end of the nutritional intervention were decreased by 39% from baseline in adults and by 23% in seniors. Moreover, the nutritional intervention increased the level of vitamin D. In fact, it significantly increased after 12 days from 29 ± 7.3 µg/L (baseline) to 46 ± 8.5 µg/L in the adult biofortified group (Figure 4B). Meanwhile, the increase in vitamin D was from 28.4 ± 7.1 µg/L (baseline) to 44 ± 10 µg/L in the seniors biofortified group. Calcium, phosphate, calcitonin and potassium levels did not differ between baseline and time 1 (Figure 4C–F). The concentrations detected were within the normal range. In the control and Mo-tablet groups, there were no changes in the bone metabolism markers at time 1 compared to baseline (Figure 4). The comparison between the groups (control vs. biofortified and Mo-tablet vs. biofortified) showed that there were significant changes in the PTH and vitamin D in the Mo-biofortified group compared to the control group and the Mo-tablet group in both adults and seniors (Figure 4A,B). Calcium, calcitonin, phosphorus and potassium were not affected by the nutritional intervention (Figure 4C–F).

## 4. Discussion

New requests in food and human nutrition are arising, with the goals of ensuring well-being, longevity and sustainability of resources, especially in aging. Thus, innovative nutritional interventions in the field are required. In the present study, we investigated the impact of the micronutrient molybdenum provided through a vegetable matrix in a cohort of adult and senior participants.

Before this work, no studies had investigated the effect of a Mo-biofortified crop on bone homeostasis in the human population. A previous study, conducted in rabbits, showed that molybdenum supplementation of the diet increased the deposition of calcium in the femur bone by 22% [35], highlighting the importance of dietary molybdenum supplementation for optimal bone metabolism.

We used a vegetable matrix (lettuce) as a carrier to supplement the diet with molybdenum. Our previous studies suggested a vegetable matrix was a good carrier for the mineral because the concentration of Mo was within the physiological range in serum and urine [28]. Thus, in the present study, we investigated the effect on bone metabolism and tested the effect with respect to a Molybdenum tablet supplementation group.

Biofortification using vegetable matrix (lettuce) was chosen because vegetables are consumed by most of the population. In this set up, we analyzed the effect on bone homeostasis and metabolism in the cohort of participants, and the results were compared within and between the groups. We found that consumption of 100 g of Mo-biofortified lettuce was able to improve bone homeostasis in the groups, while supplementation of molybdenum in tablets for 12 days did not affect the bone markers. The crop, consumed by the study population every day in a quantity of 100 g per day, contained 0.21 mg of sodium molybdate for the control group and 8 mg of sodium molybdate for the Mo-biofortified group. The tablet of molybdenum contained 150 µg of sodium molybdate. This quantity of molybdenum was chosen because 45 μg/day is the recommended dietary allowance of molybdenum [26] but deficiencies occur at low intakes, while excesses are eliminated at high intakes. An average intake of 65 µg/day of molybdenum was proposed for adults by the EFSA, while a tolerable upper intake level (UL) of 0.9 mg/kg body weight per day [36] has shown no adverse effect because it is used up in the body. This is very important because it reduces the risks of toxicity or deficiency, and molybdenum is fast eliminated in the urine [37]. Therefore, we provided a quantity of molybdenum that was safe (in line with the biomonitoring equivalent values fixed to protect against deficits and nutritional toxicity, and very distant from 31 μg/L, which is the biomonitoring equivalent toxicity value reported for plasma and serum) [27]. The serum level of molybdenum was 5 μg/L at baseline [38] and reached 6 μg/L with the intake of 150 µg of sodium molybdate in tablet form [39] and 7 μg/L with the intake of biofortified lettuce for 12 days [28].

Scientists agree that the nutritional recommendations on micronutrients should be adjusted to improve the metabolism [40,41]. This is necessary, for example, to reduce mitochondrial decay, a hallmark of aging, related to many disease processes. The idea that higher doses than the current recommendation may be necessary for preserving body homeostasis of specific tissue, such as bone tissue in the elderly, is supported by studies in the field, showing that the elderly can benefit from increased quantities of micronutrients, for example, calcium and vitamin D [42]. In the present study, a micronutrient’s effect was observed in reductions in circulating markers of bone resorption, CTX and PTH, indicating slowing of bone resorption in response to the mineral supplementation of Mo-biofortified lettuce, but not when participants were given molybdenum supplied in tablet form. Moreover, the nutritional intervention affected vitamin D levels. Vitamin D can influence bone homeostasis, both directly and indirectly. Vitamin D regulates serum calcium and phosphate levels for optimal skeletal mineralization. It influences calcium absorption and reabsorption from bone and the kidneys [43]. The mineral supplementation with the Mo-biofortified lettuce increased vitamin D levels in adults and seniors, confirming the ability of the nutritional intervention to impact two of the main regulators of bone metabolism, vitamin D and PTH. Vitamin D exerts a negative feedback on PTH [44]. Presently, we do not know the exact mechanism of action. The effect of consumption of Mo-biofortified lettuce on bone metabolism could be indirect by modulation of the gut peptide GIP and PYY levels, which are known to inhibit bone resorption [29,32,45]. However, considering the molybdenum function as an essential cofactor for mitochondrial bone physiology [46], it is possible to hypothesize that consumption of Mo-biofortified lettuce boosts the mitochondrial function and reduces osteoclast activity. Mitochondrial function is essential in the maintenance of osteoblasts’ and osteoclasts’ activity in bone. Mitochondrial dysfunction not only impairs osteogenesis but also increases osteoclast activity [47]. This accelerates age-related bone loss. Therefore, consumption of molybdenum through biofortification for 12 days could lead to optimal mitochondrial function that ameliorates bone mineral homeostasis.

In conclusion, this study showed that mineral supplementation with Mo-biofortified lettuce (8 mg/day) for 12 days in a cohort of adult and senior participants reduced bone resorption and PTH and increased vitamin D levels. Molybdenum supplementation using one tablet a day (150 ug/day) of Mo did not show any influence on the same bone markers during the intervention. The results are encouraging for preservation of bone health, especially in the elderly.

## Figures and Tables

**Figure 1 nutrients-15-01022-f001:**
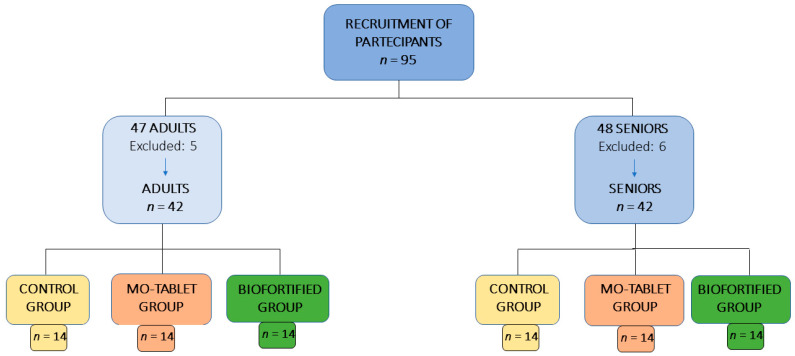
Process of participants’ recruitment in the study.

**Figure 2 nutrients-15-01022-f002:**
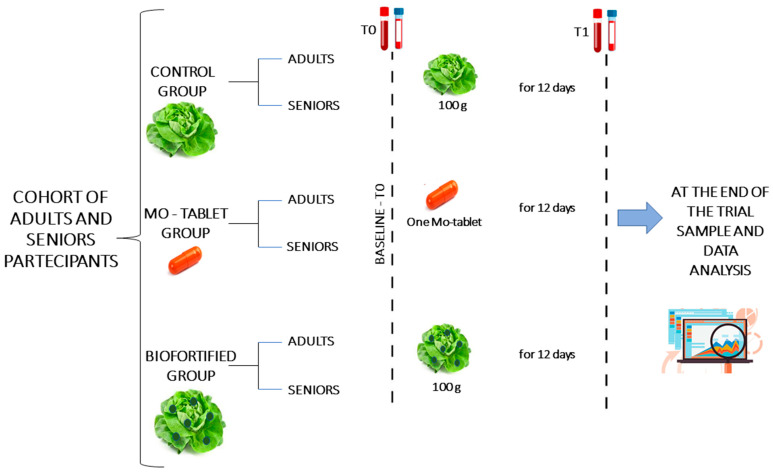
Overview of the experimental design. Samples of blood were taken at baseline and after 12 days. The control group consumed 100 g of lettuce containing 0.21 mg of sodium molybdate, and the intervention group consumed 100 g of Mo-biofortified lettuce containing 8 mg of sodium molybdate every day for a total of 12 days. The Mo-tablet group was supplied with one tablet of Mo containing 150 µg of sodium molybdate every day for a total of 12 days. Biofortification of lettuce with molybdenum was used, with a dose of 3 µmol Mo L^−1^ supplied in foliar spray as sodium molybdate (Na_2_MoO_4_) during the period of growth [17].

**Figure 3 nutrients-15-01022-f003:**
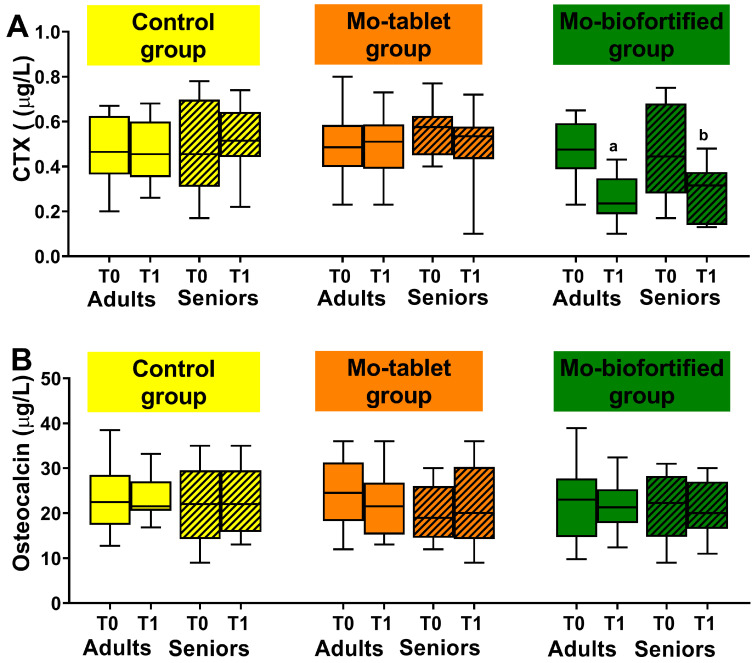
Bone remodeling markers were measured at baseline and after 12 days in the three groups: the control group (supplemented once a day with 100 g a day of lettuce), the Mo-tablet group (supplemented once a day with molybdenum in tablet form) and the Mo-biofortified group (supplemented with 100 g a day of molybdenum-biofortified lettuce). (**A**) Box plot showing serum concentration of the marker of bone resorption, CTX, in control, Mo-tablet and Mo-biofortified groups. (**B**) Box plot showing serum concentration of the marker of bone formation, osteocalcin, in control, Mo-tablet and Mo-biofortified groups. The letter a denotes a significant difference (*p* < 0.05) of Mo-biofortified lettuce compared with the control (T0 and T1), Mo tablet (T0 and T1) and T0 of Mo-biofortified lettuce in the adult group. The letter b denotes a significant difference (*p* < 0.05) of Mo-biofortified lettuce compared with the control (T0 and T1), Mo tablet (T0 and T1) and T0 of Mo-biofortified lettuce in the senior group.

**Figure 4 nutrients-15-01022-f004:**
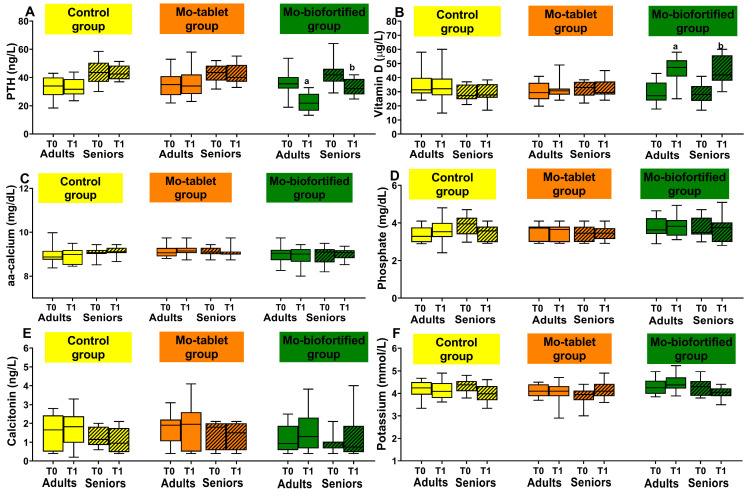
Bone metabolism markers were measured at baseline and after 12 days in the three groups: the control group (supplemented with 100 g a day of lettuce), the Mo-tablet group (supplemented once a day with molybdenum in tablet form) and the Mo-biofortified group (supplemented with 100 g a day of molybdenum-enriched lettuce). (**A**) Box plot showing concentration of PTH in control, Mo-tablet and Mo-biofortified groups. (**B**) Box plot showing concentration of vitamin D in control, Mo-tablet and Mo-biofortified groups. (**C**) Box plot showing concentration of aa calcium in control, Mo-tablet and Mo-biofortified groups. (**D**) Box plot showing concentration of phosphate in control, Mo-tablet and Mo-biofortified groups. (**E**) Box plot of calcitonin in control, Mo-tablet and Mo-biofortified groups. (**F**) Box plot showing concentration of potassium in control, Mo-tablet and Mo-biofortified groups. The letter a denotes a significant difference (*p* < 0.05) of Mo-biofortified lettuce compared with the control (T0 and T1), Mo tablet (T0 and T1) and T0 of Mo-biofortified lettuce in the adult group. The letter b denotes a significant difference (*p* < 0.05) of Mo-biofortified lettuce compared with the control (T0 and T1), Mo tablet (T0 and T1) and T0 of Mo-biofortified lettuce in the senior group.

**Table 1 nutrients-15-01022-t001:** Characteristics of the study participants defined by eligibility criteria.

Inclusion Criteria	Exclusion Criteria
Age 21–77	Presence of chronic disease
Italian ethnicity	Use of medication or dietary supplements
Body mass index 19–30 kg/m^2^	Pregnancy
Absence of chronic disease	Breastfeeding
Medical history of absence of fractures	Fractures

**Table 2 nutrients-15-01022-t002:** Characteristics of participants at the beginning of the study (baseline, T0). Values are indicated as means ± standard deviations (SD).

Participant Data	Adults (17 Females, 25 Males) *n* = 14 Subjects in Each Group				Seniors (18 Females, 24 Males) *n* = 14 Subjects in Each Group			
	Control Group Females *n* = 5	Mo-Tablet Group Females *n* = 6	Mo-Biofortified Group Females *n* = 6	*p*-Value	Control Group Females *n* = 6	Mo-Tablet Group Females *n* = 6	Mo-Biofortified Group Females *n* = 6	*p*-Value
	Mean ± SD	Mean ± SD	Mean ± SD		Mean ± SD	Mean ± SD	Mean ± SD	
Age (years)	34.6 ± 12	37.9 ± 11.9	36.6 ± 9.8	n.s.	61.8 ± 5.4	61.3 ± 5.1	61.7 ± 5.3	n.s.
Weight (kg)	69.4 ± 11	69.1 ± 10.6	69.5 ± 12.5	n.s.	70.7 ± 9.2	71 ± 5.6	74.6 ± 7.5	n.s.
Height (cm)	167.6 ± 11	164.1 ± 9.2	172.7 ± 7.7	n.s.	163.1 ± 7	164.1 ± 7.4	165.1 ± 7.2	n.s.
BMI (kg/m^2^)	24.9 ± 4.9	25.7 ± 4	23.2 ± 3.2	n.s.	26.7 ± 3.8	26.5 ± 2.9	27.4 ± 2.4	n.s.
Fat mass (%)	24.2 ± 2.1	24.1 ± 2.2	25.4 ± 6.3	n.s.	29.7 ± 3.9	28.3 ± 2	28.1 ± 2	n.s.
Lean mass (%)	75.8 ± 2.1	75.9 ± 2.2	74.5 ± 6.3	n.s.	70.3 ± 3.9	71.7 ± 2	71.8 ± 2	n.s.

## Data Availability

Data available on request due to privacy/ethical restrictions.

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
