# Peer review of "The Role of Consumption of Molybdenum Biofortified Crops in Bone Homeostasis and Healthy Aging"

_nutrients, 2023, doi:10.3390/nu15041022_

Round 1

Reviewer 1 Report

This article “The role of consumption of molybdenum biofortified crops on bone homeostasis and healthy aging” aimed to analyze the effects of consumption of biofortified crops with the micronutrient molybdenum (Mo) in bone remodeling and metabolism in a population of adults and seniors. The work is interesting with adequate data and in line with readers' interests of nutrients. However, there are still some shortcomings that need to be improved or explained.

Comments:

Section 1: Abstract

Q1. Senile bone loss is a long-term process, why the experiment period was 12 days. After 12 days of administration, will there be drug-induced changes in the indicators.

Section 2: Introduction

Q2. Osteoporosis is an age-related disease that consists in reduced bone mass, disruption of bone architecture, and increased risk of fractures. How is osteoporosis represented at the molecular level? Which constituent compounds increase or decrease? If possible, the relevant demonstrations are suggested to be supplemented.

Section 3: Results

Q3. The resolution of all the graphs needs to be improved. Please replace all figures.

Q4. What are the differences between Mo-tablet group (supplemented once a day with molybdenum in tablet) and Mo-biofortified group (supplemented with 100 grams a day of molybdenum enriched lettuce)? Did people receive equivalent content of Mo? If so, then what are the reasons that inducing the difference between these group, such as Figure 4 B?

Section 4: Discussion

Q5. Whether the safe dose of Mo is considered, whether Mo is used up in the body or can be recycled trace elements. The relevant expressions are suggested to be added.

Q6. In the conclusion part, it is recommended not to cite references, but to effectively summarize the results of this paper.

Author Response

Thank you very much for your comments and suggestions that will help us to improve the quality of the manuscript.

Comments:

Section 1: Abstract

Q1. Senile bone loss is a long-term process, why the experiment period was 12 days. After 12 days of administration, will there be drug-induced changes in the indicators.

A1. We chose an experimental period of 12 days because the bone remodeling marker levels change acutely following food intake. They are influenced in the short term (Henriksen, D.B.; Alexandersen, P.; Bjarnason, N.H.; Vilsbøll, T.; Hartmann, B.; Henriksen, E.E.G.; Byrjalsen, I.; Krarup, T.; Holst, J.J.; Christiansen, C. Role of Gastrointestinal Hormones in Postprandial Reduction of Bone Resorption. J. Bone Miner. Res. 2003, 18, 2180–2189). Therefore, a period of 12 days allows us to determine if there are drug-induced changes in the indicators. We added the sentence in the abstract, lines 28-30.

“We chose an experimental period of 12 days because the bone remodeling marker levels are influenced in the short term. Therefore, a period of 12 days allows us to determine if there are changes in the indicators.”

Section 2: Introduction

Q2. Osteoporosis is an age-related disease that consists in reduced bone mass, disruption of bone architecture, and increased risk of fractures. How is osteoporosis represented at the molecular level? Which constituent compounds increase or decrease? If possible, the relevant demonstrations are suggested to be supplemented.

A2. As you suggested we added in the introduction section (Lines 58-64) the relevant information of molecules, signals and pathway involved at molecular level in osteoporosis:

“At molecular biology level, deregulation in signals and pathway of Wnt, RUNX2, RANKL, OPG, Osx, CBFB, BMP-2, FoxOs, Nrf2, Gsα, sclerostin, lead to osteoporosis (Gao, Y.; Patil, S.; Jia, J. The Development of Molecular Biology of Osteoporosis. Int J Mol Sci 2021, 22, doi:10.3390/ijms22158182). For example, changes at genetic and epigenetic level that might lower or over-activating the Wnt-signaling pathway cause osteoporosis. Also, the mutations in LRP5 cause the osteoporosis-pseudoglioma syndrome (OPPG) autosomal disorder (Gong, Y.; Slee, R.B.; Fukai, N.; Rawadi, G.; Roman-Roman, S.; Reginato, A.M.; Wang, H.; Cundy, T.; Glorieux, F.H.; Lev, D.; et al. LDL receptor-related protein 5 (LRP5) affects bone accrual and eye development. Cell 2001, 107, 513-523, doi:10.1016/s0092-8674(01)00571-2.) Thus, the Wnt-signalling pathway is a candidate for therapeutic intervention of osteoporosis (Baron, R.; Gori, F. Targeting WNT signaling in the treatment of osteoporosis. Curr Opin Pharmacol 2018, 40, 134-141, doi:10.1016/j.coph.2018.04.011.)”

Section 3: Results

Q3. The resolution of all the graphs needs to be improved. Please replace all figures.

A 3. As you suggested we have increased the resolution of the figures and replaced all the figures. We did not see it in our copy and we are very sorry for it. Our apology.

Q4. What are the differences between Mo-tablet group (supplemented once a day with molybdenum in tablet) and Mo-biofortified group (supplemented with 100 grams a day of molybdenum enriched lettuce)? Did people receive equivalent content of Mo? If so, then what are the reasons that inducing the difference between these group, such as Figure 4 B?

A4. Thank you for your question. The Mo-tablet group was supplemented with a tablet of Mo containing 150 µg of sodium molybdate while the Mo-biofortified group was supplemented with 100 grams a day of molybdenum enriched lettuce containing 8 mg of sodium molybdate. They received different content of Mo and this accounts for differences in the results as in figure 4B.   We added this information also in the results (lines 233-236). 

“The Mo-tablet group was supplemented with a tablet of Mo containing 150 µg of sodium molybdate while the Mo-biofortified group was supplemented with 100 grams a day of molybdenum enriched lettuce containing 8 mg of sodium molybdate”

Section 4: Discussion

Q5. Whether the safe dose of Mo is considered, whether Mo is used up in the body or can be recycled trace elements. The relevant expressions are suggested to be added.

A5. As you suggested the relevant expression were added (lines 321 and 324)

Q6. In the conclusion part, it is recommended not to cite references, but to effectively summarize the results of this paper.

A6. The conclusion part was adjusted to summarize the results and the references were deleted (lines 357-362)

“In conclusion, the study shows that the mineral supplementation with the Mo-biofortified lettuce (8 mg/day) for 12 days in a cohort of adult and senior participants reduces bone resorption and PTH and increases vitamin D level. Molybdenum supplementation using 1 tablet a day (150ug/day) of Mo seems do not show any influence on the same bone markers during the intervention study protocol. The results are encouraging for preservation of bone health especially in elderly. “

Reviewer 2 Report

This randomized clinical trial (RCT) investigated the effect of mineral supplementation with Mo-biofortified lettuce on bone health in a cohort of adult and senior participants.  Results from this RCT are promising and open new directions for large sample size studies with demographic variations.  The study is well-done pilot proof of concept with proper experiments, which justifies publication in nutrients. However, a few aspects below will need to be clarified.

Have you looked for safety labs after supplements?

Is there any side effect of supplementation?

How is the dose of supplementation decided?

Table two formatting needs to be organized.   

The figure's quality can be improved since it looks blurred.

Author Response

Thank you very much for your comments and suggestions.

Have you looked for safety labs after supplements?

Thank you for this interesting question. The laboratory that produces the Mo-tablet (zeinPharma) report claim of the very highest quality and optimal purity. They report to do frequent swab tests on rooms, machines and employees' hands ensure hygiene standards of the very highest quality.

Is there any side effect of supplementation?

No, subjects did not report any side effects and we did not observe any side effects. They joined and completed the short-term intervention study in good health with best compliance and without drop-out.

How is the dose of supplementation decided?

Thanks for the question. We conducted a previous study showing that the highest molybdenum leaf tissue concentration was detected in plants biofortified with a dosage of 3.0 µmol molybdenum L-1 (0.55 mg g-1 of dry weight) supplied by foliar spray (Sabatino, L.; Consentino, B.B.; Rouphael, Y.; De Pasquale, C.; Iapichino, G.; D’Anna, F.; La Bella, S. Protein Hydrolysates and Mo-Biofortification Interactively Modulate Plant Performance and Quality of ‘Canasta’ Lettuce Grown in a Protected Environment. Agronomy 2021, 11, 1023, doi:10.3390/agronomy11061023). Thus, we decided to treated plants from plots with 3.0 µmol molybdenum L-1 . Then we measured the amount of molybdenum in control lettuce that was 0.21 mg/100 g fresh weight while the amount of molybdenum in biofortified lettuce was 8 mg/100 g fresh weight. Because the Tolerable  Upper  Intake  Level  (UL) has shown no adverse effect at the dosage of 0.9 mg/kg body weight per day (EFSA Panel on Dietetic Products, N.; Allergies. Scientific Opinion on Dietary Reference Values for molybdenum. EFSA Journal 2013, 11, 3333, doi:https://doi.org/10.2903/j.efsa.2013.3333) and 100 grams of lettuce/day are well tolerated by the subjects we decided to use the dosage of 8 mg/100 g fresh weight of Mo enriched lattuce for the nutritional intervention.

Table two formatting needs to be organized.   

As you suggested table two was formatted.

The figure's quality can be improved since it looks blurred.

Thanks, we have improved figure’s quality.